# Effectiveness, Safety and Choroidal Changes of a Fovea-Sparing Technique for the Treatment of Chronic Central Serous Chorioretinopathy with Yellow Subthreshold Laser

**DOI:** 10.3390/jcm12031127

**Published:** 2023-01-31

**Authors:** Beatriz Torrellas, Alejandro Filloy, Lihteh Wu, Jay Chhablani, Pedro Romero-Aroca

**Affiliations:** 1Ophtalmology Department, Joan XXIII University Hospital, 43007 Tarragona, Spain; 2Institut d’Investigació Sanitaria Pere Virgili (IISPV), Rovira & Virgili University, 43204 Reus, Spain; 3Clínica Oftalmològica de Tarragona (COT), 43001 Tarragona, Spain; 4Asociados de Mácula, Vítreo y Retina de Costa Rica, San José 10102, Costa Rica; 5UPMC Eye Center, University of Pittsburgh, Pittsburgh, PA 15213, USA; 6Ophthalmology Department, University Hospital Sant Joan de Reus, 43204 Reus, Spain

**Keywords:** central serous chorioretinopathy (CSC), yellow subthreshold laser, choriocapillaris, choroid, choroidal thickness, Haller layer, Sattler layer, Swept-Source optical coherence tomography, pachychoroid disease

## Abstract

The aim of this study was to evaluate the effectiveness and safety of a yellow subthreshold laser (STL) for the treatment of chronic central serous chorioretinopathy delivered in a fovea-sparing pattern and to analyze the post-laser changes in the choroidal structure by Swept-Source Optical Coherence Tomography. This study was a prospective case series of 43 eyes corresponding to 37 patients. Data were recorded at 6, 12 and 24 weeks after the STL treatment. The best-corrected visual acuity improved in 93% of the patients and remained stable in 7%. The subretinal fluid was completely reabsorbed in 27.9%, 32.6% and 69.8% of the patients at 6, 12 and 24 weeks, respectively. There were reductions in the choroidal thickness of 13.1% and 25.3% at 12 and 24 weeks, which corresponded to reductions of 17.5% and 45.9% in the choriocapillaris and Sattler layer and reductions of 12.2% and 21.2% in the Haller layer at 12 and 24 weeks, respectively (*p* < 0.05). This might account for the effect of the laser on the inner choroidal vasculature, the dysregulation of which is believed to be at the core of central serous chorioretinopathy. No laser-related complications were detected. Overall, the fovea-sparing STL was safe and effective in this series of patients.

## 1. Introduction

Central serous chorioretinopathy (CSC) is a retinal disease characterized by serous retinal detachments involving the macula, often combined with alterations in or the detachment of the retinal pigment epithelium (RPE) due to the focal leakage of fluid from the choroid [1].

CSC is the fourth most frequent retinal pathology after age-related macular degeneration, diabetic retinopathy and branch retinal vein occlusion. It is typically observed in male patients [2]. It is classified as acute or chronic based on the duration of the serous retinal detachment, which varies between different studies, ranging from 3 to 6 months. In its acute form, it is a self-limited disease which usually resolves within 4–8 weeks. However, in its recurrent or chronic form, there is a need for treatment because of the persistence of serous retinal detachment. Chronic central serous chorioretinopathy (CCSC) causes progressive RPE and photoreceptor deterioration, resulting in persistent visual impairment [3].

Optical Coherence Tomography (OCT) is an essential tool used to evaluate this disease. In recent years, new OCT techniques such as Swept-Source Optical Coherence Tomography (SS-OCT) have been developed, showing improvements in both definition and penetration and enabling the study of the choroid’s layers [4]. This has led to the new classification of the pachychoroid disease spectrum, which includes CSC. Increased choroidal thickness is understood to be caused primarily by dilatation in Haller’s layer, while there is also a thinning of the choriocapillaris and Sattler’s layer. However, our understanding of the exact pathological mechanism of this disease is still lacking [5].

Nowadays, photodynamic therapy (PDT), subthreshold laser (STL) photocoagulation and various forms of laser photocoagulation are available as treatments. STL consists of a series of repetitive laser pulses with a cooling interval, which enables the precise control of its thermal effect. By inducing the production of antiexudative factors (especially heat shock proteins, among the other identified molecules [6]) by the RPE and, probably, Müller cells, STL promotes the recovery and restoration of the outer blood retinal layer, resulting in the resorption of the subretinal fluid [7].

Although an international consensus has been met in the recently published guidelines for STL indications and application [8], registered clinical trials are still required in order to validate it. Most of the current literature favors the use of STL in widely confluent patterns so as to stimulate a high enough number of RPE cells to produce a clinically significant response.

The purpose of this study was to evaluate the effectiveness and safety of yellow (577 nm) STL for the treatment of CCSC delivered in a fovea-sparing pattern and to analyze the presence of post-laser changes in the choroidal structure of SS-OCT.

## 2. Materials and Methods

### 2.1. Study Design

The study was conducted at the University Hospital Joan XXIII of Tarragona (Tarragona, Spain) from January 2018 to December 2021 as a prospective case series of patients affected by CCSC. The study was conducted in accordance with the Declaration of Helsinki.

### 2.2. Patients

The diagnosis of CCSC was based on clinical examination, SS-OCT, Fundus Autofluorescence (AF) and Fluorescein Angiography (FA). Images were taken through SS-OCT (Triton, Topcon, Tokyo, Japan).

Patients with a history of steroid treatment in the previous 6 months as well as patients with other evolving ocular conditions were excluded. The exclusion criteria included the following: acute CSC (defined as a first episode shorter than 4 months), high myopia, amblyopia, any previous diagnosis of diabetic retinopathy, retinal vein or arterial occlusion, age-related macular degeneration, choroidal neovascularization choroidal neovascularization correlated with pachychoroid disease, uveitis or any other pachychoroid disease other than CSC.

### 2.3. Procedures

The enrolled patients received treatment with a 577 nm subthreshold laser (Easyret, Quantel medical, Courgnon d’Auvergne, France) at a power titrated to one third of the minimum energy required to produce a minimally visible burn in the macular periphery. The spot size employed was 160 microns delivered in a 5% duty cycle in a confluent pattern guided by OCT, AF or FA. The treated zone was the leakage point or points on the FA and/or the areas or hyperfluorescence on the AF, as well as the whole area with subretinal fluid (SRF) on OCT. The laser was delivered following a “fovea-sparing” principle. That is, after localizing the fovea through direct observation and retinographic correlation, the fovea itself and its surrounding 500 microns were excluded from treatment. The same experienced surgeon performed the treatment in all cases.

Medical history and demographic characteristics were documented for each patient at the baseline visit. Data were recorded for both eyes prior to the STL treatment, at 6, 12 and 24 weeks, and thereafter on a basis determined by individual evolution. We registered best-corrected visual acuity (BCVA) using Snellen visual acuity ratios, SS-OCT (macular cube and an HD line through the fovea), FA and AF. Every case was analyzed for changes in the BCVA, SRF and choroidal thickness (CT) as a whole, as well as Haller’s and Sattler’s layers, respectively. Laser-induced changes on funduscopy, OCT or AF, as well the occurrence as choroidal neovascularization, were also registered.

### 2.4. Outcomes

The primary outcome of the study was SRF absorption. SRF was measured from the neurosensory retina to the underlying RPE in the area of maximum height of SRF. Complete SRF reabsorption was defined as the absence of SRF on SS-OCT, whereas a partial reabsorption or increase in SRF was defined as a decrease or increase in the SRF thickness of at least 25%.

The secondary outcomes were CT, BCVA and the presence of adverse events.

CT was defined as the distance between the base of the RPE and the choroid-sclera junction, as displayed on SS-OCT. Haller’s layer was defined as the outer choroidal layer containing a 100 µm or larger vessel [9]. Additionally, the choriocapillaris and Sattler’s (C-S) layer were defined as the difference between the CT and Haller’s layer thickness (Figure 1).

Measurements of the CT and Haller’s and C-S layers were taken just under the area of maximum height of SRF (which was also the center of the laser-treated area), as well as 1000 µm nasal and temporal to that point. The mean of these three values produced the resulting values.

The measurements of the CT and Haller’s and C-S layers were performed by two independent masked retinal specialists at every visit to study the interobserver reliability of the data.

### 2.5. Control Group

For the control group, we selected the fellow non-treated eye in the patients with unilateral CCSC.

### 2.6. Statistical Analysis

The statistical analysis was performed using StataCorp. 2020, Stata Statistical Software: V16.1. The results are reported as the mean ± standard deviation (SD). Student’s *t*-test was used to determine the significance of the differences in the mean values. The chi-square test was used to determine the significance of the differences in the categorical variables. The intraclass correlation coefficient (ICC) was used to study the interobserver reliability of the data collected by the two independent investigators. A *p*-value of 0.05 or lower was considered statistically significant.

## 3. Results

The study included 43 eyes corresponding to 37 patients (Figure 2). The mean age was 55 ± 10.2 years old (range 37–77). The patients were mostly men (28, 75.7%), and we assessed 21 right eyes and 22 left eyes. Bilateral disease was observed in seven patients (18.9%).

### 3.1. Subretinal Fluid (SRF)

A reduction in SRF was achieved in 76.7%, 88.4% and 95.4% of the patients at 6, 12 and 24 weeks, respectively. A subgroup analysis was performed in which complete absorption of the SRF was observed in 27.9% of the patients at 6 weeks, 32.6% at 12 weeks and 69.8% at 24 weeks. There was an increase in SRF in 23.2%, 10.8% and 4.6% of the patients at 6, 12 and 24 weeks, respectively (Figure 3).

A subgroup analysis was performed to compare the patients who experienced episodes of a duration shorter than 12 months (26.67 ± 7.1 weeks, 12 patients) with the patients with long standing disease (236.97 ± 182.5 weeks, 21 patients). In the first group, SRF was completely reabsorbed in 75% of the patients after 6 months of follow-up, while 67.7% of the patients in the second group had complete reabsorption of SRF after 6 months (*p* = 0.639).

The recurrence of SRF to any degree was observed in 58.1% of the patients. Therefore, 44.2% and 13.9% of the patients required two and three STL treatments, respectively, while a single STL treatment was sufficient for 41.9% of the patients.

### 3.2. Best-Corrected Visual Acuity (BCVA)

The mean BCVA before STL was 20/50 ± 20/100, which increased to 20/33 ± 20/100 at 6 weeks (*p* < 0.05) and to 20/28 ± 20/100 after 12 weeks (*p* < 0.05). After the STL treatment, the BCVA improved in 93% of the patients and remained stable in 7%, while no patients showed deterioration.

### 3.3. Choroidal Thickness (CT)

The mean CT before STL was 374.9 ± 81.1 µm. At 12 weeks, it was 325.8 ± 59.2 µm (13.1% *p* < 0.05). During this same period, the C-S layer thickness decreased from 62.8 ± 23.9 µm to 51.8 ± 18.3 µm (17.5% *p* < 0.05), while the Haller layer thickness decreased from 312.1 ± 71.3 µm to 274 ± 55.1 µm (12.2% *p* < 0.05). The ratio between the average Haller and C-S layer thickness was 5.5 before the treatment, increasing to 7.8 (*p* < 0.05) at week 12.

The measurements after 24 weeks were as follows: mean CT 280 ± 114.6 µm (25.3% *p* < 0.05), with the C-S layer thickness decreasing to 34 ± 20.8 µm (45.9% *p* < 0.05). Additionally, the Haller layer thickness decreased to 246 ± 115.9 µm (21.2% *p* < 0.05) (Table 1).

The control group comprised the fellow non-treated eyes of 30 patients (81.1%) with unilateral disease. The mean age was 55.3 ± 12.1 years old, and 21 patients were men (76.7%). No significant differences were found in the same measurements for the control group (Table 2).

### 3.4. Adverse Events

All the OCT slides were reviewed after treatment, together with the AF and FA images. No laser-induced changes or choroidal neovascularization were found in the OCT, AF and FA images in any case.

## 4. Discussion

In this case series, the treatment with yellow STL applied through a fovea-sparing technique with 1/3 power titration yielded positive results for the treatment of CCSC, both anatomically in the OCT and functionally in the BCVA. The effectiveness of STL has already been confirmed in previous series that provided similar results in regard to the rate of disease improvement [10,11]. Most of the previously published series were shorter than that assessed in this study, and to the best of our knowledge, a fovea-sparing technique has not been previously described in relation to CCSC. Moreover, we did not find any previous reports studying the effects of STL on the CT and its different layers.

The changes observed in the CT after treatment consisted of a significant thinning at the level of both the C-S layer and Haller’s layer, which was proportionally more prominent in the former. This decrease in thickness was statistically significant from the first follow-up visit at 12 weeks. This might reflect an effect of the laser on the inner choroidal vasculature, the dysregulation of which is believed to be at the core of CCSC.

The results obtained in our study are consistent with those described in the study of Flores-Moreno et al., which demonstrated a reduction in the CT at the expense of the C-S layer in patients diagnosed with CCSC and treated with half-fluence PDT [12]. The effects on the C-S layer induced by both forms of treatment for CCSC suggest that the origin of this disease could, at least partly, be ascribed to the choriocapillaris and Sattler layer. This supports the hypothesis proposed in a recent study on the regulation of the choroidal vessel endothelial cells by Shubert C et al., who concluded that the expression of cadherin 5, the major molecule that enables cell-to-cell adhesion in the choroidal endothelium, is downregulated by corticosteroids [13]. The involvement of a glucocorticoid response pathway in CSC was demonstrated in a study conducted by Zhao et al. [14]. This may increase the permeability of the choroidal vasculature, leading to fluid leakage in the retina.

With knowledge of the pathology of CSCC and its mechanism of response to STL, various hypotheses can be proposed regarding the changes in the choroid after STL treatment. On the one hand, these changes might be due to the restoration of the integrity of the vascular capillary function, which would result in a decrease in the accumulation of interstitial fluid. On the other hand, the decrease in the vascular caliber could help to improve the structural integrity of the blood vessel wall [15,16].

Since May 2020, the supply of verteporfin (Visudyne, Cheplapharm Arzneimittel GmbH, Greifswald, Germany) has been interrupted due to a reduction in the manufacturing capacity of a factory situated in the United States, which is the single producer of verteporfin worldwide [17]. For this reason, the European Medicines Agency expects the quantity of verteporfin to be limited until the end of 2023 [18]. This global shortage of verteporfin has resulted in the need for an effective alternative to PDT, which is the main treatment option for CCSC together with STL. PDT has been compared to STL previously, with the two most recent studies being the PLACE trial [19,20] and the study of the Pan-American Collaborative Retina Study Group (PACORES) [21]. The PLACE study yielded superior results for PDT compared to STL in terms of the SRF resolution at three months. On this basis, it has been established as a reference study used to determine the superiority of PDT. However, its analysis of the results and its methodology have been subject to controversy, mostly related to the STL administration, which did not follow the methodology generally accepted by the community of STL practitioners [7], especially the need for large treatments (in the sense of treating a large surface area). The study of PACORES yielded much more positive results for STL, providing results indicating no significant differences in either anatomical or functional success between PDT and STL. A recent meta-analysis on PDT vs. STL for CCSC [22] including data from these studies, among others, also concluded that STL is a viable alternative as a first-line treatment for CCSC due to its equivalent results.

Few researchers have used a control group in the analysis of a treatment for CCSC. In our study, we included the fellow non-treated eyes, which did not display changes in their CT after STL, as opposed to the main group. This supports the hypothesis of a local effect of the laser on the inner choroidal vasculature.

There were no complications related to STL after 6 months of follow-up, such as visible laser scars, outer retinal anomalies on SS-OCT or RPE alterations on AF (Figure 4). Moreover, the results of STL treatment do not seem to be diminished by the use of a fovea-sparing technique [23]. In a previous publication by our study group [24], the fovea-sparing technique also yielded positive results in the treatment of diabetic macular edema. The results of the present study lend to support this concept of refraining from transfoveal treatment, despite this technique having delivered positive results in terms of safety when performed by experts [25]. We believe that this technique is of special interest for novice laser practitioners due to its enhanced safety, which may be useful when handling with first cases.

The main limitations of our study were the relatively small sample size of the study and the lack of a non-laser treatment control group. Ideally, multicentric large studies are required to confirm our findings.

## 5. Conclusions

To sum up, STL was an effective treatment for CCSC both functionally (in terms of the BCVA) and anatomically (in the reduction of the SRF and CT) in this case series. Its effect on the reduction in CT, especially the C-S layer, indicates the effect of the laser on the inner choroidal vasculature, which can enhance our understanding of its pathophysiology. The absence of adverse events without an apparent decrease in effectiveness supports the use of a fovea-sparing technique.

## Figures and Tables

**Figure 1 jcm-12-01127-f001:**
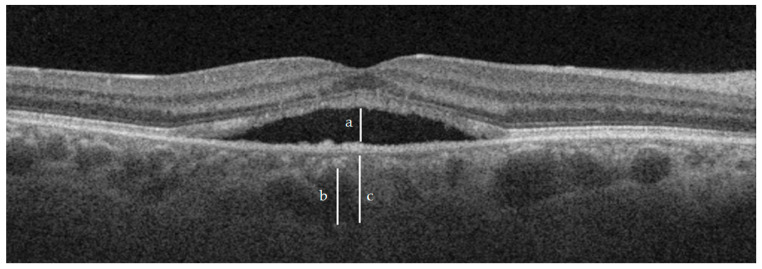
Measurements of SRF (**a**), Haller layer (**b**) and CT (**c**). For reference purposes, the Haller layer was not measured exactly under the maximum height of SRF.

**Figure 2 jcm-12-01127-f002:**
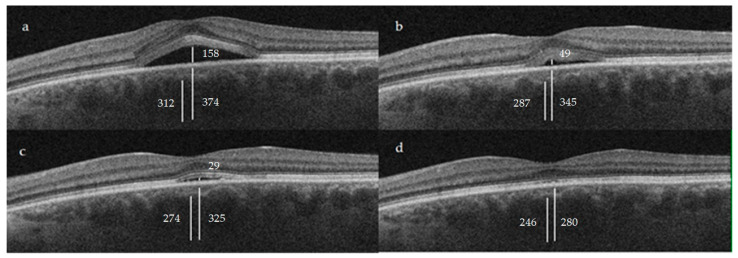
Clinical case of a 46-year-old male patient with a 6-month episode of CCSC and an initial BCVA of 20/50 at baseline (**a**). After STL treatment, we observed a progressive improvement throughout weeks 6 (**b**), 12 (**c**) and 24 (**d**), with a final BCVA of 20/20. In the image the CT, Haller and SRF measures can be appreciated, measured in µm. See Figure 1 for referencing purposes.

**Figure 3 jcm-12-01127-f003:**
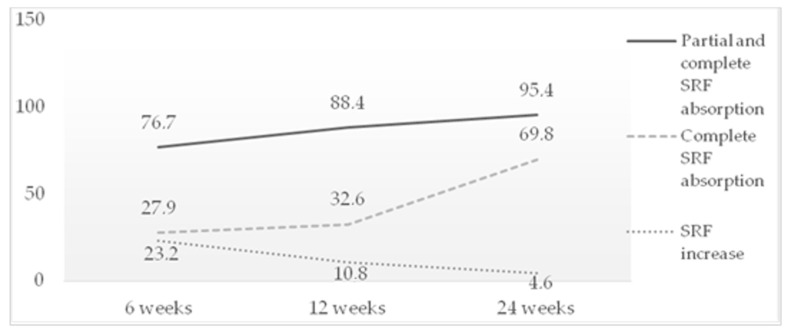
Rate of SRF absorption after STL treatment.

**Figure 4 jcm-12-01127-f004:**
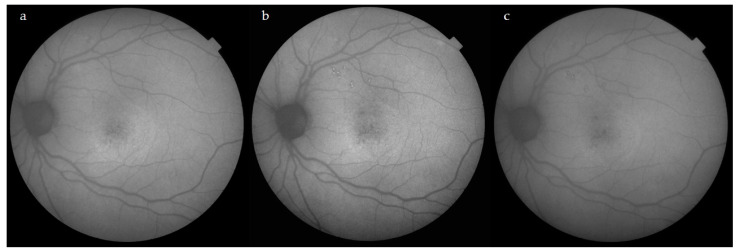
Evolution of AF. Images prior STL treatment (**a**), at 6 weeks (**b**) and at 24 weeks after STL treatment. Both (**b**,**c**) display discrete hyperautofluorescent laser spots near the superotemporal arcades due to the laser titration procedure. The area encompassing the neurosensorial detachment in the central macula does not show any laser-induced changes after receiving a total of 460 spots at the titrated power (see methods section for power titration procedure).

**Table 1 jcm-12-01127-t001:** Mean values of the CT, Haller layer and C-S layer measured in µm.

Layer	Baseline *	6 Weeks *	*p*	12 Weeks *	*p*	24 Weeks *	*p*
CT	374.9 ± 81.1	345.3 ± 71.5	0.0566	325.8 ± 59.2	<0.05	280 ± 114.6	<0.05
Haller layer	312.1 ± 71.3	287.4 ± 64.3	0.0953	274 ± 55.1	<0.05	246 ± 115.9	<0.05
C-S	62.8 ± 23.9	57.93 ± 20.8	0.3164	51.8 ± 18.3	<0.05	34 ± 20.8	<0.05

* Mean ± standard deviation (SD).

**Table 2 jcm-12-01127-t002:** Mean values of CT, Haller layer and C-S layer measured in µm in the control group.

	Baseline *	6 Weeks *	*p*	12 Weeks *	*p*	24 Weeks *	*p*
CT	293.0 ± 85.2	275.3 ± 73.4	0.3923	273.7 ± 65.0	0.9291	241.2 ± 113.8	0.1731
Haller layer	250.6 ± 75.4	238.4 ± 67.8	0.5125	236.9 ± 62.4	0.4464	245.8 ± 61.2	0.7876
C-S	42.4 ± 19.8	36.8 ± 12.4	0.1944	36.7 ± 13.4	0.1968	34.4 ± 7.4	0.0571

* Mean ± SD.

## Data Availability

Not applicable.

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
