# Peer review of "Effectiveness, Safety and Choroidal Changes of a Fovea-Sparing Technique for the Treatment of Chronic Central Serous Chorioretinopathy with Yellow Subthreshold Laser"

_jcm, 2023, doi:10.3390/jcm12031127_

Round 1

Reviewer 1 Report

This manuscript describes outcome of yellow subthreshold laser in eyes with chronic CSC. It requires significant revision in order to make it useful to readers.

1 Methods section (lines 110-111, 118-121,123-125): Please use illustration to show SRF, CT and Haller layer measurements.

2 Results section (Figure 1): The choroid-sclera junction in these figures shown by the authors are not clear. Please show the values to specify the quantitative measurements.

3 Results section (lines 200): Please add AF or IR images before and after subthreshold laser to show safety of yellow subthreshold laser.

4 Results section: The treated zone varied greatly in this study (Methods: line 95-97). Please analyze this factor.  Is the rate of complete resolution of SRF associated with the different treated zone?

5 This study lacks non laser treatment control group. Please specify this in the limitation part.

Author Response

Dear Reviewer,

Thank you for your comments, I sincerely appreciate your suggestions. I will specify the answer to your comments below.

Point 1: Methods section (lines 110-111, 118-121,123-125): Please use illustration to show SRF, CT and Haller layer measurements.

Response 1: We have added a new figure (figure 1) to better illustrate the SRF, CT and Haller measurements.

Point 2: Results section (Figure 1): The choroid-sclera junction in these figures shown by the authors are not clear. Please show the values to specify the quantitative measurements.

Response 2: We have added to the image the values of CT, Haller and SRF in figure 2.

Point 3: Results section (lines 200): Please add AF or IR images before and after subthreshold laser to show safety of yellow subthreshold laser.

Response 3: We have added a new figure (figure 4) to better illustrate the post-laser changes in AF.

Point 4: Results section: The treated zone varied greatly in this study (Methods: line 95-97). Please analyze this factor.  Is the rate of complete resolution of SRF associated with the different treated zone?

Response 4: Indeed, the treated area varies depending on the size of the leakage area. However, we have not observed changes in the treatment between different sized areas. The time of evolution of the disease seems to be the main factor influencing the degree of response. Detecting a different response depending on the size of the treated area would be very interesting towards the establishment of treatment guidelines.

Point 5: This study lacks non laser treatment control group. Please specify this in the limitation part.

Response 5: We have added the lack of a control group to the list of study limitations in line 307.

Before I conclude, I would like to thank you again for your suggestions and comments to our paper, which contribute in great measure to enhance it.

Best wishes,

Beatriz Torrellas

Reviewer 2 Report

In this paper the Authors aimed to valuate the effectiveness and safety of yellow subthreshold laser (STL) for the treatment of chronic central serous chorioretinopathy delivered in a fovea- sparing pattern and to analyze the after-laser changes in the choroidal structure on Swept-Source Optical Coherence Tomography.  This paper is based on may be of interest, I have minor concerns:

1.    The paper should be slightly revised to improve its writing.

2.    I would suggest to better explain the point of choroidal neovascularization in the exclusion criteria paragraph.  The authors should specify that “CNV correlated with pachychoroid disease” were excluded.

3.    The authors should better explain the foveal sparing technique.

4.    The global storage of verteporfin….I would suggest to better explain this paragraph. It is confused.

Author Response

Dear Reviewer,

Thank you for your comments, I sincerely appreciate your suggestions. I will specify the answer to your comments below.

Point 1: The paper should be slightly revised to improve its writing.

Response 1: In response to your suggestion, the manuscript has been submitted to MDPI for English editing, which has been of great help to enhance the quality of the paper.

Point 2: I would suggest to better explain the point of choroidal neovascularization in the exclusion criteria paragraph.  The authors should specify that “CNV correlated with pachychoroid disease” were excluded.

Response 2: We added the specification you suggested and we agree it better explains the exclusion criteria, which needed to be specified.

Point 3: The authors should better explain the foveal sparing technique.

Response 3: We used section 2, “Materials and Methods”, subsection 2.2., “Patients”, to better explain the foveal-sparing technique. We hope it is more understandable this way.

Point 4: The global storage of verteporfin….I would suggest to better explain this paragraph. It is confused.

Response 4:  In behalf of this point, we have made an exhaustive review of the paragraph and made a few changes to better explain the information contained in it.

Before I conclude, I would like to thank you again for your suggestions and comment to our paper, which contribute in great measure to enhance it.

Best wishes,

Beatriz Torrellas
